

# Evaluation of *rabi* season sesame productivity from graded nutrient doses and tillage regimes in rice fallows of southern plateau and hills region of the Indian sub-continent

Harisudan Chandrasekaran[1,*], K. Ramesh[2,*], Praduman Yadav[2], Ratnakumar Pasala[2], Elamathi Sathiah[1], Pandiyan Indiragandhi[1], Veeramani Perumal[1], Sivagamy Kannan[1], V. Karunakaran[1], Kathirvelan Perumal[1], Baskaran Rengasamy[1] and Subrahmaniyan Kasirajan[1]

[1] Tamil Nadu Agricultural University, Coimbatore, Tamil Nadu, India
[2] ICAR-Indian Institute of Oilseeds Research, Hyderabad, Telangana, India
* These authors contributed equally to this work.

Corresponding author
K. Ramesh, kragronomy@gmail.com

## ABSTRACT

**Background:** Only scattered information is available on the tillage and nutrient management information for the sesame crop following rice in the literature. Sesame as an edible oil yielding crop with high levels of unsaturated fatty acids has high international demand due to superior health benefits. Being a small seeded crop, it requires standard tillage and nutrient management to obtain optimum productivity under rice fallow ecologies. As a sequential crop after rice harvest, the tillage and nutrient management practices followed for the preceding rice have astounding effects on the succeeding sesame crop. To better understand and manipulate the agro ecology in the rice fallow culture, it is necessary to study the behaviour of sesame cultivars, in relation to the tillage requirements and macro nutrient factors that have a bearing on the productivity.
**Methods:** The aim of this work was to evaluate the productivity of rice fallow sesame in the southern plateau and hills regions of the Indian sub-continent (Tamil Nadu) with a hypothesis that tillage and nutrient management would immensely benefit the sesame crop. Field experiments were conducted at TNAU, Tamil Nadu Rice Research Institute, Aduturai, Tamil Nadu during 2019–2020 and 2020–2021 with tillage practices (reduced tillage, conventional tillage and zero tillage) and fertilizer doses (zero percent RDF, 25% RDF, 50% RDF, 75% RDF and 100% RDF) in a split plot design replicated thrice.
**Results:** The results have clearly indicated that the performance of rice fallow sesame was poor under zero till conditions as the sesame crop is poorly adapted leading to a yield penalty up to 68%. A total of 75% RDF has yielded statistically similar yield to that of 100% RDF to the rice fallow sesame. Further, neither the oil content nor the fatty acid composition was modified by tillage and nutrient management regimes.

## INTRODUCTION

Sesame crop is a short duration edible oilseed crop cultivated throughout the world (*Harisudan & Vincent, 2019*) with minimal inputs (*Oyeogbe et al., 2015*) in semi-arid and arid regions while in the tropical environment regarded as a residual moisture (*Pasala et al., 2021*) crop. The crop serves as a contingent crop after the harvest of long/medium duration rice in several states of India particularly in the Cauvery deltaic region of Tamil Nadu in India. High productivity in this crop is a function of tillage and nutrient management (*Santos et al., 2018*) particularly in the rice fallow ecologies where the physical soil structure has been deteriorated (*Yang et al., 2022*) due to puddling in the preceding rice crop. The extensive use of heavy machinery in rice farming brings about numerous benefits through the creation of a compact soil layer particularly to arrest the water loss through percolation and this could have a detrimental effect on the rice fallow sesame crop in general. This compaction may increase the mechanical strength of the soil but may be detrimental for the growth of the succeeding crop. A change in soil ecology from flooded rice to upland farming may result in soil physical constraints to sesame growth and establishment, besides, soil water deficit and aeration necessitating the need for tillage studies (*Ramesh et al., 2021*). To meet the demands of the rice fallow sesame crop, integrating soil nutrient status and the edaphoclimatic factors is an absolute necessity, besides, the contribution of left over nutrients derived from the preceding rice crop. Normally grain yields in the absence of nitrogen (N) fertilizer are functions of available soil N for plant use from net mineralization (*Chen et al., 2018*) in any crop. More than 75% of the rice fallow crops are seldom fertilized. It is generally believed that the sesame crop must, therefore, generally subsist on residual soil moisture or residual nutrients from fertilizer applied to a previous rice crop. Paddy soils are dominated by ammonium ions which are the major source of inorganic nitrogen for rice, are extracted by the roots *via* ammonium transporters and subsequently assimilated into the amide residue of glutamine by the reaction of glutamine synthetase in the roots (*Tabuchi et al., 2005*). Ammonium ions in the anaerobic soil layers of flooded rice results through modification in the chemistry of rice rhizosphere (*Fageria et al., 2011*). Although rice is an efficient user of both (ammonium and nitrate) forms of nitrogen (*Kronzucker et al., 2000*), since sesame can't survive under low land environments, conversion of rice land to aerobic conditions may facilitate the accumulation of nitrate (*Yang et al., 2021*) rather than ammonium ions. A full understanding soil physical and chemical properties in paddy-upland cropping systems is necessary (*Zhou et al., 2014*) for optimum sesame production. Though a major portion of lowland rice areas, is able to support a good second crop by virtue of carry-over residual soil moisture (due to heavy texture and high moisture retention), it is mostly mono-cropped. Suboptimal yields of crops following low land rice might be due to unfavourable physical conditions of the soil which inhibit crop growth and nutrient uptake in rice fallow (*Kar & Kumar, 2009*) particularly subsoil hard pan (*Yang et al., 2022*) which

is a serious lacunae due to poor soil fertility (*Yang et al., 2021*, *2022*). This is so important in the sense that the impervious subsoil paddy layers could be a deterrent for the sesame root to proliferate beyond the usual 30 cm although the roots may reach as high as 180 cm as evidenced by *Gloaguen et al. (2018)* while working with a dozen sesame genotypes. Research conducted elsewhere has also reported an improvement in the soil structure by growing of a rice fallow crop with suitable seeding and tillage methods (*Ishaq et al., 2001*; *Gangwar et al., 2006*).

Although sesame is one of the oldest domesticated plants in the world, its production is limited, because of its low yield (*Ashri, 1989*) and so any increase in sesame productivity under improved management would, therefore, have a large impact on the sustenance of rice-sesame cropping systems. While this knowledge creates a foundation for further, studies of sesame crop under rice fallow environments, it constituted a rather detached and partial body of work in the literature. Rice fallow sesame practiced in several states of the country mostly during the spring season after *rabi* rice and to a limited extent in north eastern states, is a step towards the horizontal expansion of sesame production in the country. However, its productivity remains abysmally low due to several factors as compared to sole crop and concerted research efforts needs to be focused to enhance its productivity (*Ramesh et al., 2019*, *2020*). Sesame under no tillage after rice has been considered as one of the climate smart practices for sustainable cropping intensification (*Derpsch et al., 2014*) which preserves the soil quality by reducing soil erosion (*Lal, 2001*). Certainly improving the root system has a high potential to increase crop productivity and the root biomass positively contributes to increased seed yield in sesame (*Su et al., 2019*). Although sesame is a low nutrient demanding crop, it needs to be supplied with balanced fertilizers (*Ramesh et al., 2019*). Although few studies directed on nitrogen (*Badshah et al., 2017*) as well as nitrogen-phosphorus-potassium (N-P-K) fertilization (*Shehu, 2014*) have provided positive responses, studies on sesame following rice are very scarce.

Therefore, studies on tillage requirement and nutrient management for rice fallow sesame should focus on the measurement of agronomic traits, such as plant height, nodes with capsules (number), number of branches and height to the first capsule, to robotically comprehend the impacts of tillage and nutrient management on the yield. To address this scarcity of research information regarding optimal tillage and nutrient management for sesame in a predominant rice belt of the country *viz*., Tamil Nadu, a 2-year field trial was conducted, with the main objective of determining the impact of tillage practices as well as nutrient management on growth and yield of sesame. Information from this study would ultimately help in developing comprehensive tillage recommendations for rice fallow sesame in the state of Tamil Nadu as well as similar rice growing soil ecologies for scaling up.

# MATERIALS AND METHODS

## Conducting the experiment

The experiments were conducted in the research farm of TNAU-Tamil Nadu Rice Research Institute, Aduthurai, Tamil Nadu, India geographically located at a latitude of 11°0′ N and 79°30′ E longitude with an altitude of 19.4 m above Mean Sea Level, during

February, March, April and May, during the years 2020 and 2021 after the harvest medium duration rice. On the basis of temperature, the station is classified as hyper-thermic (very hot). The centre falls in the humid tropical monsoon climate according to Koppen climate classification brings an average annual rainfall of about 1,000 mm. The region has a tropical wet and dry/savanna climate with a pronounced dry season in the high-sun months, and no cold or wet seasons (monsoon season) in the low-sun months, with an annual precipitation of 1,202 and 2,014 mm in 2020 and 2021, respectively. With an average annual rainfall of 1,150–1,250 mm mainly received during northeast monsoon, Cauvery River that brings water from retreating monsoon catchments and cauvery deltaic is a fertile rice growing region of the Peninsular India, where traditional rice cultivation dates back to more than two millennia.

According to the Thornthwaite climate classification as previously described by Thornthwaite (1948), the climate of the site is dry sub-humid and is being shifted to semi-arid as previously described by Raju et al. (2013) in the recent past. The soil of the experimental field was alluvial clay with a pH of 7.5, EC 0.3 dS/m and low, high and medium in available nitrogen, phosphorus and potassium contents of 242, 32 and 230 kg NPK/ha respectively. The average meteorological data of the cropping period for the two consecutive years 2020 and 2021 are presented in the Tables 1 and 2, respectively.

## Experimental design and treatments

Split plot design was employed in the experiment with three replicates, with the treatments arranged in a split plots scheme, wherein the main plots were assigned tillage methods (reduced, conventional and zero tillage) and in the subplots nutrient management practices (0%, 25%, 50%, 75% and 100% RDF; RDF: 35:23:23 kg NPK/ha). The total area of the experiment was 1,320 m$^2$, and each experimental plot consisted of ten rows of plants, totaling an area of 15 m$^2$ (5.0 × 3.0 m). The spacing used was 0.30 × 0.10 m, with two plants per hole, totaling 260 plants in the net plot harvest area of the experimental plot (9.66 m$^2$), and a population of 271,967 plants ha$^{-1}$.

## Experimental materials

The detailed description of crop varieties employed in the study is given below. CR 1009 sub 1, a popular rice variety was raised. VRI 3, a derivative of SVPR 1 and TKG 87 is a branched sesame variety released during 2017 for the state of Tamil Nadu and matures in 80–90 days recommended for irrigated conditions in the months of Dec–Jan and Feb–Mar sowing. The plant becomes yellow on physiological maturity and 25% of capsules open from top; if the harvest is delayed more than a week, after maturity, delayed shattering/late shattering type. Suitable for summer irrigated conditions. The special features of the variety are erect, indeterminate with profuse branching, four locules and white seeds.

## Experiment management

Rice crop cv. CR 1009 sub 1 was sown in the nursery on 19 August 2019, transplanted on 01 October 2019 and harvested on 30 January 2020 for the first year while for the second year, 20 August 2020, 02 October 2020 and 01 February 2021, respectively, as a bulk crop

Table 1 Meteorological data during the sesame growing period 2020.

| MSW | Temp °C | | | | Relative humidity (%) | | Wind speed (km/h) | Sunshine hours | Evaporation (mm) | Rainfall (mm) | Rainy days |
|---|---|---|---|---|---|---|---|---|---|---|---|
| | Max °C | | Min °C | | | | | | | | |
| | I | II | I | II | I | II | | | | | |
| 6 | 32.1 | 32.0 | 18.8 | 18.8 | 96.6 | 87.3 | 0.3 | 7.9 | 4.5 | 0.0 | 0.0 |
| 7 | 32.5 | 32.2 | 19.4 | 19.4 | 96.3 | 91.3 | 0.1 | 8.0 | 5.0 | 0.0 | 0.0 |
| 8 | 31.9 | 32.0 | 19.3 | 19.3 | 95.4 | 88.7 | 0.1 | 7.5 | 5.1 | 0.0 | 0.0 |
| 9 | 31.1 | 30.9 | 19.1 | 19.1 | 96.1 | 88.3 | 0.1 | 7.2 | 3.1 | 18.2 | 3.0 |
| 10 | 34.5 | 34.3 | 21.4 | 21.4 | 94.9 | 58.6 | 0.1 | 7.5 | 4.9 | 0.0 | 0.0 |
| 11 | 34.3 | 34.0 | 21.6 | 21.6 | 95.1 | 54.3 | 0.0 | 8.1 | 5.1 | 0.0 | 0.0 |
| 12 | 34.9 | 34.8 | 22.0 | 22.0 | 94.9 | 49.6 | 1.6 | 7.4 | 5.3 | 0.0 | 0.0 |
| 13 | 34.3 | 34.4 | 21.6 | 21.6 | 94.3 | 52.3 | 3.2 | 9.2 | 5.2 | 0.0 | 0.0 |
| 14 | 35.9 | 36.2 | 23.9 | 23.9 | 94.4 | 50.4 | 2.7 | 7.9 | 5.1 | 0.0 | 0.0 |
| 15 | 35.6 | 35.3 | 23.7 | 23.7 | 93.4 | 53.3 | 3.1 | 8.7 | 5.3 | 0.0 | 0.0 |
| 16 | 35.8 | 35.6 | 24.8 | 24.8 | 90.9 | 54.1 | 2.5 | 9.7 | 5.3 | 0.0 | 0.0 |
| 17 | 36.0 | 36.0 | 26.0 | 26.0 | 90.4 | 54.4 | 2.1 | 7.5 | 5.7 | 0.0 | 0.0 |
| 18 | 35.5 | 36.2 | 25.6 | 25.6 | 89.1 | 53.7 | 1.0 | 7.2 | 5.8 | 3.0 | 1.0 |

Note:
MSW, Meteorological Standard Week.

Table 2 Meteorological data during the sesame growing period 2021.

| MSW | Temp °C | | | | Relative humidity (%) | | Wind speed (km/h) | Sunshine hours | Evaporation (mm) | Rainfall (mm) | Rainy days |
|---|---|---|---|---|---|---|---|---|---|---|---|
| | Max °C | | Min °C | | | | | | | | |
| | I | II | I | II | I | II | | | | | |
| 6 | 30.4 | 30.3 | 18.3 | 18.3 | 95.1 | 61.6 | 6.1 | 8.2 | 2.1 | 0.0 | 0.0 |
| 7 | 31.8 | 31.8 | 17.1 | 17.1 | 92.7 | 62.4 | 5.0 | 8.6 | 2.7 | 0.0 | 0.0 |
| 8 | 32.0 | 29.9 | 20.1 | 20.1 | 94.7 | 75.7 | 5.9 | 4.0 | 1.6 | 17.5 | 3.0 |
| 9 | 32.7 | 32.5 | 18.9 | 18.9 | 94.9 | 59.6 | 3.8 | 7.5 | 2.2 | 0.0 | 0.0 |
| 10 | 33.9 | 34.1 | 20.4 | 20.4 | 95.1 | 49.3 | 5.7 | 8.5 | 2.7 | 0.0 | 0.0 |
| 11 | 34.1 | 34.0 | 19.9 | 19.9 | 90.1 | 53.0 | 5.1 | 8.2 | 3.2 | 0.0 | 0.0 |
| 12 | 34.4 | 34.5 | 20.2 | 20.2 | 94.1 | 52.7 | 4.8 | 6.9 | 4.2 | 0.0 | 0.0 |
| 13 | 35.7 | 36.4 | 22.9 | 22.9 | 93.6 | 51.6 | 5.4 | 6.7 | 4.1 | 0.0 | 0.0 |
| 14 | 36.9 | 36.8 | 23.3 | 23.3 | 92.4 | 49.6 | 5.7 | 7.3 | 3.5 | 0.0 | 0.0 |
| 15 | 35.1 | 34.2 | 22.7 | 22.7 | 92.1 | 61.6 | 3.8 | 5.5 | 3.2 | 17.6 | 1.0 |
| 16 | 35.3 | 36.0 | 24.2 | 24.2 | 92.1 | 50.7 | 4.3 | 6.4 | 4.1 | 0.0 | 0.0 |
| 17 | 36.5 | 36.5 | 24.7 | 24.7 | 92.3 | 50.6 | 3.5 | 7.9 | 4.2 | 0.0 | 0.0 |
| 18 | 36.9 | 36.7 | 24.1 | 24.1 | 89.6 | 63.6 | 4.0 | 7.6 | 4.5 | 0.0 | 0.0 |
| 19 | 34.2 | 35.7 | 24.0 | 24.0 | 84.1 | 63.1 | 4.7 | 8.1 | 4.4 | 0.0 | 0.0 |
| 20 | 36.1 | 36.3 | 23.1 | 23.1 | 86.9 | 62.3 | 5.3 | 7.5 | 4.0 | 10.1 | 1.0 |
| 21 | 35.9 | 35.7 | 22.7 | 22.7 | 83.7 | 54.7 | 7.3 | 3.4 | 3.7 | 0.0 | 0.0 |

with uniform crop management practices. A fertilizer dose of 150:50:50 kg N: $P_2O_5$:$K_2O$/ha was followed as a standard practice for the state of Tamil Nadu. While full $P_2O_5$ was applied as initial dose at the time of transplanting, parceling was carried out for N and K. N and K was parceled as 25% at transplanting, 25% at active tillering, 25% at panicle initiation stage and 25% at heading stage. No micronutrient supplementation was done for the rice crop. After the harvest of rice crop, the land remained undisturbed until the sowing of sesame crop. While all tillages were sown uniformly for the first crop (sesame cv. VRI 3) on 11 February 2020, and the second year 08 February 2021 for zero tillage and 02 March 2021 for reduced and conventional tillage due to heavy soil moisture conditions. Conventional and reduced tillage plots were sown by drawing line with help of trench hoe and rope and applying fertilizer and covering with soil. Zero tillage sown by digging soil with trench hoe with application of fertilizers. After 10–15 days of emergence, thinning was performed, leaving one plant per hole. Three light irrigations were given during both the years as the rainfall was insufficient to maintain sufficient soil moisture. Recommended blanket fertilizer dose was carried out as per the recommendations for the state of Tamil Nadu for 100% RDF (35:23:23 kg NPK/ha), while for other doses, the amount of which were decided by the treatment details. The source of N used was urea (N 46%), P as Super phosphate ($P_2O_5$ 16%) and K as muriate of potash ($K_2O$ 60%) as per the treatments discussed. Entire dose of N, P and K were applied basally at the time of sowing. Weeding and hoeing were carried out at 20 DAS to manage crop-weed competition to maintain below the Economic Threshold Level. Other crop management practices for biotic stresses were carried out as and when necessary, according to the recommendations.

## Harvest and evaluated variables

Harvest of the bulk rice crop was carried out at physiological maturity at 150 days and the seed yield was recorded at 12% seed moisture (Data not reported). The height of the sesame crop was measured from the ground level to the tip of the plant at 30, 60 and at harvest and expressed in cm. The number of branches plant$^{-1}$ was counted at 30, 60 and at harvest and expressed in no plant$^{-1}$. Harvesting of the first and second sesame crops was carried out at 83 and 85 days after sowing (harvest on 05[th] May and 24[th] May respectively), following which certain characteristics were evaluated, including: number of branches/plant, no of capsules/plant; biomass yield; seed yield; harvest index; oil content; and fatty acid composition. Seed yield was determined by weighing seeds at 12% seed moisture from the plants in the net plot area.

## Oil content

Oil content of sesame was analyzed using a bench top pulsed nuclear magnetic resonance (NMR)—Oxford-MQC-5 analyzer (London, UK), supplied with preloaded 'easy cal' software, calibrated with known oil sesame seed samples. The calibration was performed with a 40 mm diameter sample probe, 5 MHz operating frequency, four scans, 1 s recycle delay and 40.00 magnetic box temperature. NMR room temperature was maintained at 25 °C ± 2. Before construction of calibration, sample seeds were dried by keeping them at 80 °C for 8 h in a hot air oven (*Yadav & Murthy, 2016*).

## Fatty acid profiling

Hexane on a Soxhlet apparatus was used to extract oil from seeds (Extraction unit, E-816, Buchi, Flawil, Switzerland). methanolic KOH was used to trans esterify the oil at 55 °C for 30 min using 2 ml. The organic phase was extracted with hexane and thereafter washing with water to ensure neutral pH reaction. This was followed by dying on anhydrous sodium sulfate and methyl esters were obtained with concentrated nitrogen. To determine fatty acid composition an Agilent 7890B gas chromatograph (Santa Clara, CA, USA) from Agilent Technologie was used C. An initial temperature of 150 °C was maintained for the carrier gas was nitrogen which was set to a constant gas flow of 1.2 ml/min. The fatty acid composition was determined by identifying and calculating the relative peak area percentages by GC post run analysis EZChrom elite compact software as previously described by *Anjani & Yadav (2022)*.

## Statistical analysis

Analysis of variance (ANOVA): The data collected for each evaluated variable were subjected to ANOVA for split-plot design. Analysis of variance was done using SPD procedures of SAS version 9.2 (*SAS Institute, 2009*). After testing the ANOVA assumptions, treatment means were tested for significance (LSD) at 5% probability levels.

# RESULTS

## Environmental conditions and crop development

Mean atmospheric minimum temperature conditions of both years were nearly similar while 2021 was a little warmer (34.47 °C) than 2020 (34.16 °C) as evident from the mean maximum temperature for the period 06–18 MSW of first year (2020) (Table 1) and 06–21 MSW of the second year (2021) (Table 2) while mean minimum 21.66 and 22.09 °C for the corresponding period. Total precipitation during the 2021 (45.2 mm in three rainy days) was greater than in 2020 (21.2 mm in only one rainy day). While total precipitation in the second year (2021) was adequate for sesame production, in-season variability in rainfall necessitated supplemental irrigation in the first year (2020) to avoid moisture stress and yield reduction. Physiological maturity of the cultivar, was reached at 82–85 DAS during both the years. The sesame cultivar moved in normal growth phases and stages at the same time during both the years with slight change in harvest date due to management issues. Additional rainfall during 2021 forced deferment on the dates of sowing for the conventional tillage and reduced tillage and accordingly the harvest date delayed by a fortnight.

## Progressive plant height increments

In the plant height (PH), there was interaction between tillage and fertilizer dose in both the crops up to 60 DAS, while in the second crop, even 30 DAS had a significant interaction, an increase in PH as a result of an increase in RDF rate, occurred. However the plants were shorter in the first year. Significantly taller plants were observed *viz.*, 75 and 80.5 cm for the conventional tillage (Table 3) during first and second year, respectively. In RDF, taller plants were observed when 100% RDF was applied to the crop *i.e.*, 69.1 and

**Table 3  Plant height increments during 2020 and 2021.**

| Treatment | 2020 | | | 2021 | | |
|---|---|---|---|---|---|---|
| | 30 DAS | 60 DAS | Harvest | 30–60 | 60 DAS | Harvest |
| **Tillage practice** | | | | | | |
| Reduced tillage | 16.3 | 32.6 | 66.9 | 17.2 | 33.9 | 69.6 |
| Conventional tillage | 16.8 | 33.6 | 75.0 | 17.8 | 34.0 | 80.5 |
| Zero tillage | 16.0 | 33.2 | 51.1 | 17.0 | 30.5 | 52.8 |
| CD ($P = 0.05$) | 0.44 | NS | 0.94 | 0.49 | 1.31 | 1.04 |
| **Fertilizer dose** | | | | | | |
| Control | 15.9 | 32.0 | 59.8 | 16.9 | 31.6 | 62.8 |
| 25% RDF | 16.6 | 33.3 | 62.1 | 17.6 | 34.5 | 65.4 |
| 50% RDF | 15.9 | 31.8 | 64.5 | 16.7 | 28.7 | 67.8 |
| 75% RDF | 16.4 | 32.8 | 66.2 | 17.2 | 33.5 | 69.5 |
| 100% RDF | 17.0 | 34.8 | 69.1 | 18.0 | 35.8 | 72.6 |
| CD ($P = 0.05$) | 0.56 | 1.12 | 1.39 | 0.63 | 1.10 | 1.42 |
| **Interaction** | | | | | | |
| A × B | 0.98 | 2.12 | NS | 1.09 | 2.08 | NS |
| B × A | NS | 2.17 | NS | NS | 2.13 | NS |

72.6 cm respectively for first and second year, respectively and all sub optimal doses of nutrients proved to be inferior.

## Branches per plant and SPAD values

In the branch development, tillage and fertilizer dose interacted without temporal variation and it was observed that, as the RDF rate increased, more number of branches have developed. The number of branches/plant outnumbered 2020 probably due to higher availability of soil moisture and the proportionate higher availability of nutrients (N, P and K) to the sesame plants that was consequently extracted from the soil due to even spread of rainfall during 2021. However the branching was shy during the first year as compared to the second year (2021) except zero tillage management over the first year (2020). Higher branched sesame plants were observed 4.1 and 6.3 plant$^{-1}$ in the conventional tillage during first and the second year, respectively. In regards to nutrient management, unlike tillage management in both the years 100% RDF, had the maximum number of branches plant$^{-1}$ (2.66 and 4.90 for the first and second year, respectively) and all sub optimal doses of nutrients proved to be inferior (Table 4).

SPAD reading is an indirect measurement of Chlorophyll concentration in plants which is the most important photosynthetic pigment for capturing light and driving electron transport in reaction centres. The SPAD reading is closely correlated with leaf chlorophyll content. At 60 DAS, conventional tillage has recorded higher values during both the years of study (34 and 44.5 during 2020 and 2021, respectively). Similarly, application of 100% recommended doses of fertilisers have recorded the highest SPAD reading as compared to

Table 4 Branches per plant and SPAD value at 60 DAS.

| Treatment | SPAD value | | Branches/plant | |
|---|---|---|---|---|
| | 2020 | 2021 | 2020 | 2021 |
| **Tillage practice** | | | | |
| Reduced tillage | 33.90 | 36.60 | 3.50 | 5.70 |
| Conventional tillage | 34.00 | 44.50 | 4.11 | 6.30 |
| Zero tillage | 30.50 | 31.80 | 2.66 | 4.90 |
| CD ($P = 0.05$) | 1.31 | 1.31 | 0.13 | 0.13 |
| **Fertilizer dose** | | | | |
| Control | 28.70 | 30.80 | 3.00 | 5.20 |
| 25% RDF | 31.60 | 35.70 | 3.26 | 5.50 |
| 50% RDF | 33.50 | 38.60 | 3.45 | 5.70 |
| 75% RDF | 34.50 | 39.80 | 3.63 | 5.80 |
| 100% RDF | 35.80 | 43.20 | 3.78 | 6.00 |
| CD ($P = 0.05$) | 1.10 | 1.97 | 0.12 | 0.17 |
| **Interaction** | | | | |
| A × B | 2.08 | 3.53 | 0.22 | 0.21 |
| B × A | 2.13 | 3.31 | 0.21 | 0.22 |

other sub optimal doses of RDF during the same period (35.8 and 43.2 respectively for 2020 and 2021).

## Number of capsules/plant

Sesame yield components include number of plants per unit area, number of branches per plant, number of capsules per leaf axil, seeds per capsule and seed weight (*Delgado & Yermanos, 1975*). To pinpoint the most important factor that determines sesame yield, we have recorded the number of capsules/plant in different treatments. In number of capsules/plant (NC), it was detected that, conventional tillage has recorded higher number of capsules (49.3 and 52.3 during 2020 and 2021, respectively) as compared to zero tillage and reduced tillage during both the years. Further, as the RDF rate increased, so did NC. The maximum values were obtained at 100% RDF in the year 2021 than 2020. However, more NC values (45.7 and 48.8) were obtained during 2021 than 2020 as compared to other sub optimal doses.

## Seed yield

Productivity was marginally higher in the second crop across the tillage and nutrient management regimes obviously due to additional soil moisture in the soil profile due to three spells of rains aiding in sesame agronomic performance (Table 5). While the capsule number/plant was enhanced by 6–8%, the seed yield increase was just 2–4%. Among tillage regimes, conventional tillage, wherein tilling the soil two times followed by bringing the soil to fine tilth has resulted in the higher yields (477 and 492 kg ha$^{-1}$ during 2020 and 2021, respectively) as compared to either reduced tillage (408 and 425 kg ha$^{-1}$ during 2020

**Table 5 Sesame yield and yield attributing characters as influenced by tillage and fertilizer doses under rice fallow ecologies.**

| Treatment | No of capsules plant$^{-1}$ | | Productivity (kg ha$^{-1}$) | | Biomass (kg ha$^{-1}$) | | Harvest index | |
|---|---|---|---|---|---|---|---|---|
| | 2020 | 2021 | 2020 | 2021 | 2020 | 2021 | 2020 | 2021 |
| **Tillage practice** | | | | | | | | |
| Reduced tillage | 43.0 | 46.1 | 408 | 425 | 2,918 | 3,059 | 14.0 | 13.9 |
| Conventional tillage | 49.3 | 52.3 | 477 | 492 | 3,453 | 3,567 | 13.8 | 13.8 |
| Zero tillage | 34.1 | 37.1 | 334 | 343 | 2,396 | 2,442 | 13.9 | 14.0 |
| CD ($P$ = 0.05) | 1.20 | 1.29 | 65.4 | 73.2 | 119 | 124 | | |
| **Fertilizer dose** | | | | | | | | |
| Control | 38.7 | 41.8 | 366 | 363 | 2,623 | 2,606 | 14.0 | 13.9 |
| 25% RDF | 40.3 | 43.3 | 390 | 408 | 2,828 | 2,932 | 13.8 | 13.9 |
| 50% RDF | 42.3 | 45.3 | 406 | 424 | 2,925 | 3,067 | 13.9 | 13.8 |
| 75% RDF | 43.6 | 46.7 | 422 | 440 | 3,007 | 3,169 | 14.0 | 13.9 |
| 100% RDF | 45.7 | 48.8 | 447 | 465 | 3,229 | 3,340 | 13.8 | 13.9 |
| CD ($P$ = 0.05) | 0.76 | 0.86 | 27.5 | 42.7 | 99 | 102 | | |
| **Interaction** | | | | | | | | |
| A × B | 1.72 | 1.83 | NS | NS | 187 | 194 | – | – |
| B × A | 1.57 | 1.67 | NS | NS | 193 | 199 | – | – |

Note:
    SEm for No of capsules/plant. B:0.29; Productivity. B:9.3 (for the year 2021).

and 2021, respectively) or zero tillage (334 and 343 kg ha$^{-1}$ during 2020 and 2021, respectively) for rice fallow sesame. While every additional 25% of NPK has recorded an additional 3–5% enhancement in yield, application of 25% NPK over control has recorded a quantum jump of 11% increase in yield during the second year. The same trend followed for biomass yield and harvest index (Table 5).

## Sesame oil and fatty acid composition

The oil content of sesame remained unaffected due to tillage practices as well as nutrient doses in the present study (Table 6). The results have exhibited 14.3–15.4% saturated fatty acid while 82.2–86.3% unsaturated fatty acids. The fatty acid composition has revealed that among the unsaturated fatty acids, oleic acid, a monounsaturated omega-9 fatty acid (47.3–50.6%) was the dominant fatty acid followed by linoleic acid (C18:2, omega-6 unsaturated fatty acid; 37.7–38.9%). On the other hand, palmitic acid (8.01–9.0%) was the dominant fatty acid followed by stearic acid (3.9–4.58%) among the saturated fatty acids.

## DISCUSSION

Mean atmospheric minimum temperature conditions of both years were nearly similar while 2021 was a little warmer (34.47 °C) than 2020 (34.16 °C). Additional rainfall during 2021 might have forced deferment of the dates of sowing for the conventional tillage and reduced tillage and accordingly the harvest date delayed by a fortnight. In the PH, as the RDF rate increased, an increase in PH occurred due to the higher availability of nutrients

**Table 6 Sesame seed oil content and fatty acid composition as influenced by tillage practices and fertilizer management.**

| Treatment | Oil content (%) | | Palmitic acid (%) | | Stearic acid (%) | | Oleic acid (%) | | Linoleic acid (%) | |
|---|---|---|---|---|---|---|---|---|---|---|
| | 2020 | 2021 | 2020 | 2021 | 2020 | 2021 | 2020 | 2021 | 2020 | 2021 |
| **Tillage practice** | | | | | | | | | | |
| Reduced tillage | 43.4 | 43.1 | 8.92 | 8.85 | 4.52 | 4.49 | 48.1 | 48.01 | 37.5 | 37.31 |
| Conventional tillage | 45.2 | 45.1 | 8.01 | 7.96 | 3.93 | 3.92 | 50.1 | 50.03 | 37.2 | 37.21 |
| Zero tillage | 44.9 | 44.7 | 8.83 | 8.79 | 4.58 | 4.48 | 47.5 | 47.39 | 38.4 | 38.35 |
| CD ($P = 0.05$) | NS | NS | NS | NS | NS | NS | NS | NS | NS | NS |
| **Fertilizer dose** | | | | | | | | | | |
| Control | 44.5 | 44.2 | 7.8 | 7.8 | 4.0 | 3.9 | 49.7 | 49.5 | 38.0 | 38.0 |
| 25% RDF | 44.8 | 44.9 | 9.2 | 8.9 | 4.7 | 4.7 | 46.5 | 46.5 | 38.5 | 38.4 |
| 50% RDF | 43.0 | 42.8 | 9.0 | 9.0 | 4.2 | 4.2 | 46.9 | 46.7 | 38.9 | 38.6 |
| 75% RDF | 45.0 | 44.9 | 8.3 | 8.3 | 4.1 | 4.0 | 49.3 | 49.3 | 37.8 | 37.7 |
| 100% RDF | 45.0 | 44.9 | 8.7 | 8.7 | 4.7 | 4.7 | 50.6 | 50.4 | 35.6 | 35.4 |
| CD ($P = 0.05$) | NS | NS | NS | NS | NS | NS | NS | NS | NS | NS |

(N, P and K) to the sesame plants that were extracted from the soil coupled with availability of sufficient soil moisture. It was observed that different rates of NPK might have maintained higher soil nutritional status of the sesame crop.

The number of branches/plant outnumbered 2020 probably due to higher availability of soil moisture and the proportionate higher availability of nutrients (N, P and K) to the sesame plants that was consequently extracted from the soil due to even spread of rainfall during 2021. The SPAD reading is closely correlated with leaf chlorophyll content and the values recorded at 60 DAS have shown that conventional tillage has recorded the highest value during both the years of study (34 during 2020 and 44.5 during 2021). Similarly, application of 100% recommended doses of fertilisers have recorded the highest SPAD reading as compared to other sub optimal doses of RDF during the same period (35.8 and 43.2, respectively, for 2020 and 2021). Since sesame demands all essential nutrients balanced fertilization is one strategy for high productivity (*Ramesh, Patra & Biswas, 2017*; *Ramesh et al., 2019*, *2020*). Of which nitrogen (N) is a key constituent in chlorophyll structure, requires a sufficient supply of nitrogen for dry matter production, and consequently seed yield (*Ramesh et al., 2021*). Nitrogenous fertilizers were reported to improve the leaf chlorophyll content of sesame (*Nosheen et al., 2019*) earlier also.

One of the important goals in sesame management is to ensure high capsule density. Higher number of capsules/plant is one of the ways for increasing sesame seed yield as it ensures extra number of capsules/leaf axil (*Baydar, Marquard & Turgut, 1999*) and as a result capsule density/plant is increased. This advantage was realized in the second year since extra capsule setting ability/axil in sesame is an important advantage in the effort to increase the per plant seed yield (*Baydar, 2005*).

But simply having high capsule density/plant was not good enough to enhanced yields as evident from second year yield data. According to our data, the main yield attributing

character of sesame seems to be the number of capsules/plant as reported by *Baydar (2005)*. Theoretically, if a plant provide more number of capsules/plant, more capsules per unit area are then acquired and accordingly more seed yield might be provided. However, in the present study although there was an increase in the capsule number per plant in the second year, it didn't translate into yield. Probably the late formed capsules didn't get sufficient time to improve the seed filling which need to be studied in depth. The behaviour of NC indicated that an increased rate of RDF, may provide greater availability of N, P and K in the soil, allowed greater translocation of nutrients to the sesame plants, remobilized into the capsules, consequently might have promoted an increase in the productivity of the sesame. The differences in capsule number/plant might influence seed number and size too. There were temporal variations in sesame for tillage and nutrient response either in the absence of fertilization (0 kg ha$^{-1}$ NPK) or 100% RDF ostensibly due to moisture availability in the crop root zone.

Seed yield was higher in the second crop, probably due to climatic variables, such as temperature, relative humidity, and rainfall interfering with sesame agronomic performance (Table 5). Currently there are great deal of information is available in regards to temperature response of sesame.

As expected, tillage has improved sesame yield, as reported (*Alemayehu et al., 2023*) and rice-sesame cropping system is no exception. Recently, *Yunyan et al. (2023)* have found that a decrease in temperature, hampered root length, shoot length and fresh weight of sesame at early seedling stage with significant effect at 18 °C. In the current study, the minimum temperature immediately after sowing were well above the base temperature of 10 °C and thereafter a steady increase favoring sesame productivity under the conventional tillage regimes which might have interfered with root system architecture of sesame to realise high seed yield over other tillage regimes. The meteorological conditions were favourable enough for the sesame crop during both years and the results are in conformity with those obtained by *Fageria (1998)*, who also reported that climatic variables are likely to influence NPK fertilization efficiency, and tillage to determine the yielding capacity of any plant.

Among tillage methods, conventional tillage wherein tilling the soil for two times followed by bringing the soil to fine tilth has resulted in the higher yields as compared to either reduced tillage or zero tillage for rice fallow sesame. Weeds would have competitive advantage under zero tillage due to reduced water availability, and intensify the crop-weed competition pressure (*Ramesh, Patra & Biswas, 2017*). Destruction of soil structure in the surface soil and subsoil hardpans from intensive tillage for rice (*Ogunremi, Lal & Babalola, 1986*) needs to be broken to make the field suitable for sesame establishment and plant stand. Although, no tillage ensures sustainable cropping intensification (*Derpsch et al., 2014*) through preservation of soil quality (*Lal, 2001*), the requirements of the sesame crop couldn't be met through zero tillage regimes. Crop stand establishment of sesame is considered as very important for sesame production which is in jeopardy when tillage is foregone or kept at a minimum scale. In the initial two fortnights after sowing sesame exhibits a relatively deferred aboveground biomass development (*Amare, 2011*) to an extent of 35 DAS (*Ribeiro et al., 2018*), particularly tailored by tillage regimes. Since,

sesame crop's early root development and proliferation are expected to be controlled by soil fertility (*Gloaguen et al., 2022*) application of nutrients might have certainly benefited the crop the most. Our results are in conformity to the findings of *Uzun et al. (2012)*, who reported that in spite of higher energy savings and lower land preparation costs due to no-till for sesame, there was yield penalty too.

It has been unclear whether the low yields of crops following rice paddies were due to altering soil physical or mineral characteristics, or both (*Yang et al., 2022*). The performance of rice fallow sesame is poor under zero till conditions as the sesame crop is poorly adapted to rice fallow regime (*Harisudan & Sapre, 2019*). Probably the soil pressure under zero till is a constraint to sesame since a soil pressure of at least 1.1 kg/cm$^2$ is beneficial for sesame production (*Gabrillides & Akritidis, 1970*).

It is very well accepted that sesame is well adapted to nutrient starved soil environments and thus, in practice, fertilization is infrequent whether sole crop or rice fallow crop. Further, the crop management practices, interactions among the soil physical and chemical factors have astounding effect on the productivity as well as use efficiency of the applied nutrients. Nitrogen becomes a limiting nutrient, since sesame is sown in rice fallows, the crop is seldom supplied with nutrients (*Ramesh et al., 2019*). The literature lacks solid fertilization recommendations or guidelines for a rice fallow sesame crop, yet there are many evidences to illustrate marginal yield gains due to N under field conditions. As the water logged condition in rice increases water soluble iron (*Gotoh & Patrick, 1974*), graded increase in fertilizer dose might have improved soil nutrient availability and higher sesame yield. The extent of the response outdone with 75% RDF application to 100% RDF. While every additional 25% of NPK has recorded an additional 3–5% enhancement in yield, application of 25% NPK over control has recorded a quantum jump of 11% increase in yield during the second year, plausibly due to additional soil moisture availability in the soil profile from three rain spells during the sesame growing season under rice fallows. Our data show that both capacity for and efficiency of rice fallow sesame production is greater for the combined application of N, P and K indicating a highly specialized requirement to the nutrients which has not hitherto not been recognized for sesame production. Given the potential importance of tillage and nutrient application to the enhancement of sesame yield, we propose the tillage and nutrient management strategy to capitalize on the high capacity of rice followed by sesame cropping system for realizing the optimal yield potential. It is clearly established that only appropriate land management practices coupled with nutrient management would ensure higher crop yields, in rice fallow sesame as well, although tillage systems are location specific.

The oil content of sesame remained unaffected due to tillage practices as well as nutrient doses in the present study (Table 6). Sesame oil is a balanced fatty acid composition with equal percentage of oleic and linoleic acids (*Liu, Zheng & Xu, 1992*) which is one of the prime indicators of the nutritional value of the sesame oil (*Gharby et al., 2012*) and particularly the oleic/linoleic fatty acids ratio of sesame makes it important for human health (*Oboulbiga et al., 2023*). The oil content and composition of sesame might be influenced by soil fertility since plants synthesize a huge variety of fatty acids *de novo* from precursors derived from photosynthates for eg. The content of oleic acids, linoleic acid,

linolenic acid, palmitic acid, and stearic acid varied between 36.13% and 43.63%, 39.13% and 46.38%, 0.28% and 0.4%, 8.19% and 10.26%, and 4.63% and 6.35%, respectively (*Kurt, 2018*). It was postulated that trade-off between oil and protein may be the regulatory mechanism for their negative response to high nitrogen levels in oilseed crops like sunflower and soybean (*Chen et al., 1999*). Therefore, maintaining adequate supplies of Nitrogen besides other macro nutrients in the soil would improve crop productivity and quality. Application agricultural inputs was found to modify the oleic acid, linoleic and linolenic acid composition in the monoculture rapeseeds (*Stepien, Wojtkowiak & Pietrzak-Fiecko, 2017*). In general, the variations in seed yield and fatty acid profile corresponded well with growing season precipitation and temperatures in the given environment (*Obour et al., 2017*). In the current study, since sesame is rotated with rice, the deleterious effects of mono cropping might have been nullified and the treatment variables could not provide any effect on the fatty acid composition of sesame. Our results are in conformity with the findings of *Priya et al. (2022)* who couldn't notice any significant changes in sesame fatty acid composition due to tillage and fertilizer management.

## CONCLUSION

It is generally believed that the sesame crop establish well after the harvest of the rice crop with residual nutrients from fertilizer applied to a previous rice crop. Preparatory tillage for the sesame crop is indeed an absolute necessity from the trials conducted. It could be concluded that rice fallows of the deltaic regions in the southern plateau and hills region of the Indian subcontinent can be greened with sesame with conventional tillage to ensure proper seed germination and root growth for accelerated early growth to ensure early crop growth in the rice fallow ecology. This need be combined with the recommended dose of fertilizers for rice fallow sesame crop to ensure high productivity.

## ACKNOWLEDGEMENTS

The authors acknowledge the support of TNAU, Coimbatore and ICAR-IIOR for extending necessary facilities for the conduct of the experiment.

### Funding

The authors received no funding for this work.

### Competing Interests

Ratnakumar Pasala is an Academic Editor for PeerJ.

### Author Contributions

- Harisudan Chandrasekaran conceived and designed the experiments, performed the experiments, analyzed the data, prepared figures and/or tables, and approved the final draft.
- K. Ramesh conceived and designed the experiments, performed the experiments, analyzed the data, prepared figures and/or tables, and approved the final draft.

- Praduman Yadav conceived and designed the experiments, performed the experiments, analyzed the data, prepared figures and/or tables, and approved the final draft.
- Ratnakumar Pasala performed the experiments, prepared figures and/or tables, and approved the final draft.
- Elamathi Sathiah performed the experiments, authored or reviewed drafts of the article, and approved the final draft.
- Pandiyan Indiragandhi performed the experiments, authored or reviewed drafts of the article, and approved the final draft.
- Veeramani Perumal performed the experiments, authored or reviewed drafts of the article, and approved the final draft.
- Sivagamy Kannan performed the experiments, authored or reviewed drafts of the article, and approved the final draft.
- V. Karunakaran performed the experiments, authored or reviewed drafts of the article, and approved the final draft.
- Kathirvelan Perumal performed the experiments, authored or reviewed drafts of the article, and approved the final draft.
- Baskaran Rengasamy performed the experiments, authored or reviewed drafts of the article, and approved the final draft.
- Subrahmaniyan Kasirajan performed the experiments, authored or reviewed drafts of the article, and approved the final draft.

## Data Availability

The weather data is available in the Supplemental File.

## Supplemental Information

Supplemental information for this article can be found online at http://dx.doi.org/10.7717/peerj.17867#supplemental-information.

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
