# Peer review of "Evaluation of *rabi* season sesame productivity from graded nutrient doses and tillage regimes in rice fallows of southern plateau and hills region of the Indian sub-continent"

_PeerJ, doi:10.7717/peerj.17867_

## Round 0.1 · original submission · Major Revisions

· Academic Editor

Major Revisions

Dear Dr. Ramesh,

Your work has been evaluated by 4 independent experts. All of them agreed that this work could be published in PeerJ. However, 3 of the experts believe that significant revisions are necessary before publication. Kindly review all reviewers' comments carefully, address them accordingly, and incorporate appropriate changes into the manuscript.
With best regards,

**Language Note:** The review process has identified that the English language must be improved. PeerJ can provide language editing services - please contact us at [email protected] for pricing (be sure to provide your manuscript number and title). Alternatively, you should make your own arrangements to improve the language quality and provide details in your response letter. – PeerJ Staff

Reviewer 1 ·

Basic reporting

.

Experimental design

Okay

Validity of the findings

Okay

Additional comments

Please re-write the conclusion with key findings.

·

Basic reporting

Basic reporting
At present the article is average in language and requires professional English. There are typos and grammatical errors and incomplete sentences throughout the text that requires major revision. The research article titled “Evaluation of post-rainy season sesame with graded nutrient doses and tillage regimes in rice fallows of southern plateau and hills region of the Indian sub-continent” deals with the impact of judicious fertilizer use and tillage practices on an rice fallow sesame crop. The research article is well formulated, relevant and carried out intensively which will generate scientific insight to agronomic technologies under residual soil moisture conditions in rice fallow sesame. The study is original fitting the scope of the journal. In my opinion, this research article lacks deep insight explanation overall. The article vocabulary seems average for the standards of the journal. The overall word strength of the article can be worked upon and more fluent language of vocabulary can be served. There are several minor typing errors throughout the text. The result and discussion part consists of only results and there is absence of mechanism and references supporting the claims. Therefore, I would suggest the authors to carry out a careful and extensive revision of the text and include the required mentioned data to make the article more significant and impactful.
Note: The similarity index of this article is about 35 % in text, should be reduced to below 15%.

Title: Kindly recast the title as the title is not catchy and seems lengthy.
Abstract: Abstract should be recasted with information of the findings in result section, conclusion of the research findings should be added in one or two sentences and overall improve the vocabulary as per journal standards. Keywords are missing in the text.
Introduction: Restructure the introduction part with a conceptual understanding and include more recent references pertaining to importance of tillage practices in conservation agriculture, as you just focused on nutrient management.
Material methods: Exhaustive explanation about all the parameters to be avoided, should be precise and effective. Indicate agrometerological data in figure for better representation.
Result and discussion: Discussion part is complex, should be revise to specifics.

Section Line no Comments
Abstract 24-29 Nowhere soil texture is added. Mention the experimental soil texture in methods.
31-32 yield penalty indicated compared to which tillage practices ?
32 Indicate yield values clearly for 75% and 100 % RDF.
50 “Via” should be written in italics, should be followed throughout the entire manuscript.
89 “viz” should be written in italics, followed to complete manuscript.
217 give space after 2021.
219 indicate full stop after fortnight.
229 & 338 “i.e” should be written in italics, use of italics and roman fonts for symbols in scientific text should be followed.
272-273 kg/ha in text was written kg ha-1, follow only one indication in entire manuscript.
292-294 the sentence is repeated in the results too in 217-219. Rephrase the sentence or delete the text in results section.

In tables:
32 Table 6, indicate percentage symbol clearly in all the fatty acids and also remove interaction row.
In all tables, Indicate main & sub plots clearly in the tables for better representation and readers understanding. NS indicate “non-significant”, mention abbreviations clearly in bottom of the tables.

Experimental design

The experimental design is unambiguous and relevant. The research is within the scope of the journal and will fulfill the knowledge gap.

Validity of the findings

Findings were assessed with relevant literature.

Additional comments

NA

Reviewer 3 ·

Basic reporting

Thanks for the invitation to evaluation the article 'Evaluation of post-rainy season sesame with graded nutrient doses and tillage regimes in rice fallows of southern plateau and hills region of the Indian sub-continent' for evaluation and possible publication in 'PeerJ'. I am happy to inform you that I have been able to check the whole manuscript to find out its suitability for publication and I confirm that it may be published after resolving the several issues which are listed below:
1. Abstract: The aim and treatments of the study are not clear and that should be mentioned clearly in the revised article.
2. Introduction: I suggested to incorporated latest information related to the current observation and also must be included aims of the study at the end of the section
3. M & M: All methodologies should be mentioned clearly
4. Results and Discussion: I suggest to authors to check data and related discussion based on all Tables and Figures
5. Conclusion: Authors should also add the challenges/limitations of the study in the section
Finally I recommended to publish it after the revision.

Experimental design

I am happy with experimental design and I do not have any comments. But, all methodologies should be mentioned clearly.

Validity of the findings

Results and Discussion: I suggest to authors to check data and related discussion based on all Tables and Figures

Additional comments

I am happy to inform you that I have been able to check the whole manuscript to find out its suitability for publication and I confirm that it may be published after resolving the several issues which are listed below:
1. Abstract: The aim and treatments of the study are not clear and that should be mentioned clearly in the revised article.
2. Introduction: I suggested to incorporated latest information related to the current observation and also must be included aims of the study at the end of the section
3. M & M: All methodologies should be mentioned clearly
4. Results and Discussion: I suggest to authors to check data and related discussion based on all Tables and Figures
5. Conclusion: Authors should also add the challenges/limitations of the study in the section
Finally I recommended to publish it after the revision.

Reviewer 4 ·

Basic reporting

While the article adequately presents an introduction and background illustrating the alignment of the work within the broader field of knowledge, it requires a more organized structure and clearer exposition, particularly in articulating the issues. Sufficient literature references have been cited. Several suggestions have been made for better understanding and improvement of the manuscript to meet the standard of the journal. Therefore, to accept the article a major revision is required.

Experimental design

While the submitted article effectively outlines the research questions and incorporates a pertinent statement highlighting its significance. However, clarity needs improvement. The data analysis seems less precise statistically and if possible, adding of SEm in the table will provide more clarity. There are some corrections made in the tables that also need to be thoroughly checked and addressed.

Validity of the findings

The distinctiveness and novelty of the work need to be adequately addressed, and a robust conclusion should be provided.

Additional comments

The submitted manuscript contains numerous suggestions and comments. It is recommended to address technical corrections and enhance language clarity to improve reader comprehension.

Annotated reviews are not available for download in order to protect the identity of reviewers who chose to remain anonymous.

---

## Round 0.2 · accepted · Accept

· Academic Editor

Accept

Dear Dr Ramesh,

One of the reviewers has undertaken to re-evaluate your work. He stated that this work in its current version could be published in PeerJ. Congratulations!

With best regards,

Reviewer 3 ·

Basic reporting

As per my comments, authors have resolved all issues. Therefore, the current form of article may be accepted to publish in PeerJ

Experimental design

As per my comments, authors have resolved all issues. Therefore, the current form of article may be accepted to publish in PeerJ

Validity of the findings

As per my comments, authors have resolved all issues. Therefore, the current form of article may be accepted to publish in PeerJ

Additional comments

I do not have additional comments